behaviour, cognition

cooperation, interdependence hypothesis, field experiment, alarm call, audience effect, *Pan*

**Authors for correspondence:**
Cédric Girard-Buttoz
e-mail: cedric_girard@eva.mpg.de
Roman M. Wittig
e-mail: wittig@eva.mpg.de

# Information transfer efficiency differs in wild chimpanzees and bonobos, but not social cognition

Cédric Girard-Buttoz[1], Martin Surbeck[1,2], Liran Samuni[1,2,3], Patrick Tkaczynski[1], Christophe Boesch[1], Barbara Fruth[4,5], Roman M. Wittig[1,3], Gottfried Hohmann[1] and Catherine Crockford[1,3]

[1]Department of Primatology, Max Planck Institute for Evolutionary Anthropology, Leipzig, Germany
[2]Department of Human Evolutionary Biology, Harvard University, Cambridge, MA, USA
[3]Taï Chimpanzee Project, Centre Suisse de Recherches Scientifiques, Abidjan, Ivory Coast
[4]Faculty of Science, School of Biological and Environmental Sciences, Liverpool John Moores University, Liverpool L3 3AF, UK
[5]Centre for Research and Conservation, Royal Zoological Society of Antwerp, Koningin Astridplein 20-26, B-2018 Antwerp, Belgium

  CG-B, 0000-0003-1742-4400; MS, 0000-0003-2910-2927; PT, 0000-0003-3207-2132;
RMW, 0000-0001-6490-4031; CC, 0000-0001-6597-5106

Several theories have been generated to understand the socio-cognitive mechanisms underlying the unique cooperative abilities of humans. The 'interdependence hypothesis' posits first, that the cognitive dimension of human cooperation evolved in contexts when several individuals needed to act together to achieve a common goal, like when hunting large prey. Second, the more interdependent individuals are, the more likely they are to provide services to conspecifics in other contexts. Alternatively, the 'social tolerance hypothesis' proposes that higher social tolerance allows conspecifics to cooperate more efficiently and with a wider range of partners. We conducted the first field experimental evaluation of both hypotheses in our closest living relatives by contrasting chimpanzees to the less interdependent but more tolerant bonobos. We compared each species' performance during a cooperative task: informing conspecifics about a danger. We presented Gaboon viper models to 82 individuals from five wild communities. Chimpanzees arriving late at the snake were significantly more likely to have heard a call and less likely to startle, indicating that chimpanzees were better informed about the presence of the threat than bonobos. This stems from clear species differences in how individuals adjusted their calling decisions to the level of information already available. Chimpanzees were more likely to call and produced more alarm calls when they had not yet heard a call, whereas bonobos did so when they already heard a call. Our results confirm the link between interdependence and cooperation performance. These species differences were most likely driven by differences in motivation rather than in cognitive capacities because both species tended to consider audience knowledge in their decision to call. Our results inform theories on the evolution of human cooperation by linking inter-group competition pressure and in-group cooperative motivation and/or capability.

## 1. Introduction

Humans have evolved sophisticated cooperative systems that are unprecedented in the animal kingdom and characterized by an exceptional ability to operate large scale group-level cooperation with non-relatives [1]. The interdependence hypothesis posits that the cognitive dimension underlying general human cooperative abilities evolved in food acquisition contexts in which several humans needed to coordinate their actions to acquire large prey [2]. It also

states that humans are more likely to help one another in non-collaborative contexts because they need each other in collaborative contexts [2]. The interdependence hypothesis hereby conceptually links two different forms of human cooperation: (i) 'collaboration' (also referred to as mutualism [3]) which refers to the joint action of two or more conspecifics towards a common goal that each participant could not have reached on their own; and (ii) 'non-collaborative' cooperation, which refers to a service provided by an individual (the cooperator) to a conspecific and which provides some form of fitness benefit to the recipient [4]. This link is corroborated by empirical cross-cultural studies showing that the need for successful coordinated action in a given society increases prosocial tendencies (i.e. the likelihood to engage in non-collaborative cooperation) in economical games [5] and food sharing contexts [6]. Hereafter, we define 'cooperation' as both 'collaborative' and 'non-collaborative' forms of cooperation and specify when we refer to one type in particular. The interdependence hypothesis was initially proposed to understand the evolution of unique human cognitive abilities but the theoretical foundation of this hypothesis can be extended to non-human species. Moreover, in order to fully understand the evolutionary origin of the potential link between socio-cognitive abilities, prosocial tendencies and cooperation structure, comparative studies across non-human species are essential.

Besides being our closest genetic relatives, bonobos (*Pan paniscus*) and chimpanzees (*Pan troglodytes*) are ideal species to investigate the interdependence hypothesis. These species share similar social organization (multi-male multi-female communities with a high degree of fission–fusion dynamics and female dispersal), yet they also exhibit striking differences in their level of broad-scale coordinated actions [7]. Chimpanzees are highly territorial and engage in group-level structured territorial border patrols, and hostile, sometimes lethal, inter-group encounters [8,9]. Most chimpanzee populations hunt monkeys in groups, which increases success [10,11], and some (e.g. in our study population in Taï) also engage routinely in collaborative coordinated hunting [10]. By contrast, bonobos, are not territorial, do not engage in border patrols and inter-group encounters can be peaceful [12] so that the fitness costs of not coordinating action during inter-group context in bonobos is probably lower than in chimpanzees. Bonobos at LuiKotale (our study population) also hunt monkeys [13], and hunts can involve multiple group members (B. Fruth, G. Hohmann 2017, unpublished data). Yet, whether this is a coordinated action is not known. Therefore, following the argument of the interdependence hypothesis [2], the more interdependent chimpanzees should have evolved more complex socio-cognitive skills, be more prosocial and thus more likely to engage in non-collaborative cooperation and to perform collaborative activities than bonobos.

However, the direct empirical comparisons currently available, which are entirely based on comparative tests conducted in captivity, mainly do not support this prediction. In fact, while a comparative study on gaze following and intention attribution found that bonobos performed better than chimpanzees [14], another study reported better performance of chimpanzees in perspective taking and knowledge attribution tasks [15]. In terms of prosociality, experiments show that while both species help conspecifics to obtain food, bonobos tend to be more proactive than chimpanzees in providing help (reviewed in [16]). Finally, bonobos outperformed chimpanzees in a collaborative task when the reward was highly monopolizable [17]. These later observations led to the formulation of an alternative to the interdependence hypothesis: the social tolerance hypothesis [17].

This hypothesis posits that the higher social tolerance of bonobos should allow them to cooperate more efficiently and with a wider range of partners than chimpanzees. Additional support for the social tolerance hypothesis comes from captive and field studies showing that the tolerance of bonobos allows them to spontaneously share food with out-group members [18] while chimpanzees are only willing to collaborate with specific in-group members, such as those with whom they share a close relationship [19]. However, to our knowledge, no studies have so far directly compared the socio-cognitive abilities and non-collaborative cooperative tendencies of bonobos and chimpanzees in a wild setting. This is important because the overall cooperation dynamic and related level of interdependence vary in captivity compared to the wild (e.g. apes do not need to defend a territory or acquire food collaboratively in captivity). Such differences between wild and captive settings may affect the incentive to cooperate and hence may alter the outcome of cooperative and socio-cognitive tests.

Accordingly, the aim of our study was to contrast the interdependence and the social tolerance hypotheses by comparing the performance and socio-cognitive capacities of western chimpanzees from Taï and bonobos from LuiKotale, during a non-collaborative cooperative task based on a natural context in the wild. We used a snake model stimulus paradigm allowing the apes to express a non-collaborative cooperative acts, namely to produce alarm calls that alert conspecifics to a deadly threat and hereby potentially enhancing conspecifics' fitness. Alarm calling is broadly used as an example of cooperative behaviour in vertebrates (e.g. [3,4]). The cooperative nature of alarm calls is highlighted by studies showing that birds [20] and mammals [21,22] produce alarm calls only in the presence of others. This suggests these calls function to make conspecifics aware of a threat, which is a cooperative service provided by the caller. Furthermore, in primates, some even adapt their calling behaviour to the danger level to which conspecifics are exposed [23]. It is important to note that most of these studies focus on the behaviours involved and remain agnostic as to the precise cognitive mechanisms, such as whether or not second-order intentionality or full mental state attribution of the audience's need is involved. In chimpanzees, snake model presentations revealed that alarm calls were produced while monitoring the audience [24,25], and that individuals were more likely to call and produced more alarm calls when ignorant individuals were present in the audience (i.e. individuals who had neither seen the snake nor heard an alarm call [26]). Chimpanzees thus appear to perceive, to some extent, the need for the audience to be informed and behave accordingly, although the cognitive mechanisms that this may require remain debated (see also [25]). Whether bonobos express similar strategies remains unknown as, to our knowledge, experimental studies presenting a predator model have never been conducted with this species.

Alarm calling has a number of possible functions across bird and mammal species such as confusing, deterring or mobbing the predator and/or informing conspecifics (reviewed in [27]). The function of interest in this study is the transmission of information such that the audience gains a benefit by being aware of the nearby threat, thereby reducing the danger. Vipers are a deadly threat to chimpanzees and bonobos if bitten, but they are unlikely to prey upon them.

Alarm calling at vipers has been shown to primarily function to inform other individuals in chimpanzees [25,26]. Because the function of calls produced around vipers is unknown in bonobos, we assumed that the calls also function to inform others, especially given that snake-associated behaviour is broadly comparable to that observed in chimpanzees. Informing individuals present in the vicinity of a danger can be seen as a non-collaborative cooperative task [3]. The success at this task should reflect a species' overall cooperative tendency and its socio-cognitive abilities to respond to the needs of others. We are not aiming to investigate in our study the detail of what specific socio-cognition is involved in completing the task.

The first aim of the study was to compare the efficiency of the alarm calling system (as a measure of success at solving a non-collaborative cooperative task) between wild bonobos and chimpanzees, using the same, highly salient, naturally present stimuli in both species: a model of a Gaboon viper (hereafter 'snake model'). All communities tested have been observed to see and react to these snakes within their natural habitat using various behaviours, including startles and alarm calls. Importantly, not all individuals emit alarm calls upon seeing real Gaboon vipers ([26], C. Girard-Buttoz 2016, personal observation for the bonobos). Variation in the likelihood to alarm call provides a basis for examining interspecies differences in the efficiency of the alarm calling system and the factors that affect the likelihood to call. If the interdependence hypothesis holds true, we predict that the more 'interdependent' Taï chimpanzees evolved specific cooperation-related cognitive skills (here, signalling depending on the knowledge status of the audience about the snake). Accordingly, chimpanzees are expected to be better at solving the alarm calling task (i.e. informing all other conspecifics). By contrast, the 'social tolerance hypothesis' predicts that comparatively more 'tolerant' LuiKotale bonobos are more efficient at transmitting information to the entire audience by being less selective cooperators and thus globally better at the informing task.

In order to assess performance at the task, we first used the startling reaction upon seeing the snake of individuals who were not the first to arrive at the snake as a proxy for the degree of information they possess about the snake. Specifically, we expected that informed individuals would startle less upon seeing the snake. Second, we directly evaluated the acoustic and visual information received by each individual before and upon arriving at the snake.

The second aim of the study was to understand the underlying mechanisms explaining potential species differences in information transfer efficiency by comparing the social cues that trigger alarm call production in chimpanzees and bonobos. Specifically, we assessed if potential signallers take the audiences' knowledge into account in their decision to produce alarm calls by analysing if potential signallers were more likely to call and produced more alarm calls when (i) no call had been uttered yet and (ii) some individuals who had not seen the snake (ignorant individuals) were present in the audience.

## 2. Material and methods

### (a) Study communities
We conducted the study within the Taï Chimpanzee and the LuiKotale Bonobo Projects between November 2015 and December 2017 on five fully habituated wild ape communities: three chimpanzee communities (Taï East, Taï North and Taï South) at the Taï National Park, Côte d'Ivoire and two bonobo communities (Bompusa and East) at the LuiKotale bonobo research site, Democratic Republic of Congo. For consistency, we defined for this study the same three age categories for both species. Adults were defined as all males and females greater than or equal to 10 years of age (i.e. all adults and subadults, [28]). Juveniles were defined as non-adult individuals who are not carried by their mother and thus could arrive at the snake independently from their mother. Infants were defined as individuals carried by their mother. The composition of the study communities is given in the electronic supplementary material, table S1.

### (b) Experimental protocol
On the experiment day, L.S., P.T. or an assistant followed a party of the study communities and indicated his position to the experimenter (C.G.B.) sending GPS location from device to device using Garmin Rino GPS. C.G.B. placed himself ahead of the group, out of visual range of the apes (greater than 200 m) and placed the snake model behind a log or a large root on the expected path of the party (electronic supplementary material, figure S1B). We chose experimental locations on trails or in relatively open areas allowing for optimal filming of the apes' reactions to the snake models. We aimed for similar visibility in the area around the snake in both species to avoid ecological bias resulting from differences in habitat density. All the apes approaching the snake model location were video-taped as soon as they were in visual range and until all the apes exited the area surrounding the snake. We used a three camera protocol to video-tape, in detail, the behavioural reactions of the apes to the snake model and their call production (see details in the electronic supplementary material). We also recorded continuously throughout the experiment the party composition around the snake as well as all the calls heard by individuals in other parties (see the electronic supplementary material).

We conducted a total of 33 different snake model presentations at different locations (21 in chimpanzees and 12 in bonobos). On a few occasions, the same snake model was repeatedly presented on consecutive days at the same location (see the electronic supplementary material). This allowed us to increase our sample size to 42 experiment days. After excluding the last experiment in chimpanzees for which the main camera fogged, we were left with 32 separate snake presentations and 41 experimental days.

In total, 82 adults and juveniles saw the snake model at least once (52 chimpanzees and 30 bonobos). In order to avoid habituation to the experimental protocol, we made sure to leave at least two weeks between two exposures to different snake models for each individual and conducted 23 mock trials during which all experimental conditions were met but in the absence of the snake model stimuli (see the electronic supplementary material).

### (c) Video analysis
We synchronized and coded all the videos of the experiments using Mangold Interact 14.6.0.0. Each individual entering the visual range of any video was treated as a focal. We detected when the focal saw the snake, which was clearly visible on the video because the apes systematically stopped their locomotion towards the snake, looked in the direction of the snake and either startled, stared at the snake or resumed approaching the snake at a slower pace. We also coded whether the ape startled (defined as jumping or running away from the snake in a direction not in line with general travel direction) upon detecting the snake. Finally, we coded all the alarm calls produced by the focal individual during the experiment and all the focal behaviour, its distance to the snake and its position in space when visible on the video (see the electronic supplementary material).

The identity of the caller was assessed either visually seeing lip movement in association with alarm calling on the video, or

based on the observers' speaking the identity of the caller after each call on the video. The alarm calls comprised *alarm hoos* and *alarm barks* for the chimpanzees and *soft alarm*, *alarm whistle* and *alarm barks* for the bonobos. All of these calls were produced during encounters with real vipers (see details in the electronic supplementary material and electronic supplementary material, figure S3).

## (d) Statistical analysis and model descriptions

We used a series of general linear mixed models (GLMMs) to test our predictions regarding species differences in the general efficiency of the alarm calling system, and in the socio-cognitive parameters influencing whether an individual called or not and the number of calls produced. The structure of all the models is summarized in the electronic supplementary material, table S2.

### (i) Information transfer efficiency model

In our first model (model 1a), we assessed the general efficiency of the information transfer around the snake in both species. We used the likelihood for individuals to startle when they were the first to see the snake as a baseline for the startling rate of each species. We then quantified the reduction in startling rate between the first to arrive and individuals arriving later at the snake as a proxy for the level of information about the threat that was available to these individuals prior to seeing the threat. We ran a GLMM with a binomial error structure, using 'startled upon seeing the snake, yes or no' as a response. To test whether bonobos and chimpanzees differed in how much the startle response was reduced between the first to see the snake and also all subsequent arrivers who saw the snake (hereafter late arrivers), we included an interaction in the model between the two predictors species and first arriver (yes or no). In addition, we controlled for individual sex and age class (infant, juvenile or adult), and for the number of exposures to snake models each individual had before each experiment. We included infants here because, even though they arrive on the back of their mother in the vicinity of the snake, they may walk on their own to the snake itself.

### (ii) Information available to late arriver models

We took the perspective of receivers to run two additional models to assess species differences in the level of information available to late arrivers. In the first model (model 1b), we investigated the likelihood for a late arriver to have received auditory information about the threat before approaching the snake using a GLMM with binomial error structure with 'late arriver heard an alarm call before seeing the snake: yes or no' as a response. Individuals were considered to have heard an alarm call if an alarm call was produced while they were in the party around the snake or if they could hear an alarm call while being in a neighbouring party to the one present at the snake.

In the second model (model 1c), we investigated the likelihood for a late arriver to receive visual information about the snake location while approaching the snake. We ran a GLMM with a binomial error structure and 'a conspecific was on the ground within 5 m of the snake with its body oriented towards the snake (later on defined as 'snake-oriented-body', electronic supplementary material, figure S4) yes or no' as a response. In both models, species was used as a test predictor and individual's age class and sex as control predictors.

### (iii) Triggers of alarm calling models

In a second set of models, we took the perspective of the signaller and aimed at understanding the factors explaining species differences in the information available to the late arrivers by investigating the parameters triggering alarm calling behaviour in each species. In model 2a, we assessed which socio-cognitive parameters affected the likelihood to call during the experiment. For each individual in each experiment, we split the experimental time into time slots (see the electronic supplementary material) depending on the presence or absence of individuals present in the party around the snake who had not seen the snake yet (hereafter 'ignorant individuals').

We ran a GLMM with a binomial error structure to evaluate the parameters affecting the likelihood to call for each individual in each time slot using 'call: yes or no' as a response. To test if individuals adjusted their calling behaviour depending on the knowledge level of the audience, we used two test predictors: (i) someone already produced an alarm call, yes or no (auditory information available), and (ii) are there ignorant individuals in the audience (i.e. individuals who have not seen the snake, visual information). To test if individuals considered who else could inform the audience in their decision to call, we used a third test predictor: (iii) the number of other potential signallers. We used (iv) 'order in which individual see the snake' (the first to see the snake on a given experiment day gets a one, the second to see the snake a two, etc.) to assess if individuals who see the snake later were less likely to inform conspecifics than individuals who see it first. We added all the two-way interactions between each of these four parameters and species as our aim was to assess species differences in parameters triggering the likelihood to call. We also incorporated as a test predictor the sex of the individual in interaction with species because males and females have different social status in relation to one another in each species. Finally, we added age class (i.e. adult and juvenile), number of snakes seen before each experiment and the time since the individual had seen the snake in this particular experiment as control variables (to control for the possibility of habituation, i.e. that alarm calling fades across time in each individual independently from external elements [29]). We did not include infants in the analysis here because their calling behaviour might not be representative of a fully cognitively developed individual of each species.

Finally, we used a subset of the data including only individuals which gave alarm calls to investigate which factors influenced the number of calls uttered. We fitted another GLMM with a Poisson error structure and the number of calls produced as the response variable (model 2b). The predictor and control variable structure was identical to the one of model 2a. In both models 2a and 2b, we used the log of duration of each time slot as an offset term.

In all models, to avoid pseudo replication, we incorporated four random effects, the individual identity, the study community, the snake model number and the experiment number. For all models, we included the maximal random slope structure between each fixed predictor (test and control) and each random effect. All analyses were conducted in R 3.5.1 [30] using the function *glmer* from the package 'lme4'. In each model, we tested for species difference in the test predictors by comparing the full model to a corresponding null model. We checked for model stability and the absence of collinearity and overdispersion issue (model 2b) which revealed no violation of the models' assumptions (see the electronic supplementary material).

## 3. Results

### (a) Social parameters and general response to the snake model

We conducted 41 experiments using seven different snake models on 82 juvenile and adult chimpanzees and bonobos. Detailed statistical distributions of parameters related to party size around the snake, frequency of exposure to the snake and general differences in bonobo and chimpanzee

**Table 1.** Results of models investigating the information available to late arrivers (models 1a, 1b and 1c). (s.e. indicates the standard error of the estimate for each predictor. The coded level for each categorical predictor is indicated in brackets. Control predictors are italicized. Significant p-values ($p < 0.05$) are indicated in bold. $CI_{low}$ and $CI_{high}$ indicate the lower and upper limits of the 95% confidence interval for the estimates of each predictor. The 'focal' indicates the receiver approaching the snake.)

| model | response | predictor | estimate | s.e. | $CI_{low}$ | $CI_{high}$ | $\chi^2$ | p |
|---|---|---|---|---|---|---|---|---|
| 1a | startle (Y/N) | intercept | −1.51 | 0.84 | −21.47 | 0.30 | — | — |
| | | species (chimpanzee) | −1.60 | 0.87 | −19.32 | 0.24 | — | — |
| | | first to see the snake (yes) | 1.47 | 0.87 | −0.17 | 24.33 | — | — |
| | | species: first to see the snake | 3.15 | 1.28 | 1.07 | 31.27 | 6.95 | **0.008** |
| | | *number of snake seen* | −0.53 | 0.40 | −8.36 | 0.25 | 1.07 | 0.302 |
| | | *sex of the focal (male)* | 0.78 | 0.59 | −1.13 | 8.83 | 1.50 | 0.220 |
| | | *age of the focal (infant)* | 1.01 | 1.08 | −9.40 | 14.79 | 0.97 | 0.616 |
| | | *age of the focal (juvenile)* | −0.22 | 0.66 | −6.31 | 4.96 | — | — |
| 1b | late arriver heard an alarm call before seeing the snake (Y/N) | intercept | 0.03 | 1.35 | −40.93 | 45.12 | — | — |
| | | species (chimpanzee) | 4.46 | 1.78 | 0.708 | 101.46 | 6.32 | **0.012** |
| | | *sex of the focal (male)* | 0.15 | 0.77 | −25.67 | 29.42 | 0.03 | 0.865 |
| | | *age of the focal (infant)* | −2.98 | 1.38 | −60.59 | 3.05 | 2.86 | 0.239 |
| | | *age of the focal (juvenile)* | −0.15 | 0.89 | −27.94 | 28.08 | — | — |
| 1c | a conspecific was with snake-oriented-body when the late arriver saw the snake (Y/N) | intercept | 0.28 | 0.90 | −1.88 | 2.70 | — | — |
| | | species (chimpanzee) | −0.43 | 1.01 | −3.23 | 1.80 | 0.18 | 0.675 |
| | | *sex of the focal (male)* | −0.20 | 0.46 | −1.37 | 0.92 | 0.17 | 0.676 |
| | | *age of the focal (infant)* | 1.14 | 0.93 | −1.61 | 13.20 | 1.51 | 0.470 |
| | | *age of the focal (juvenile)* | 0.26 | 0.53 | −1.08 | 1.70 | — | — |

calling behaviours around the snake are provided in the electronic supplementary material, table S3; Movies S1 and S2 show examples of chimpanzee and bonobo reactions to the snake model.

## (b) Information available to late arrivers (models 1a, 1b and 1c)

Our results from models 1a and 1b both indicate that chimpanzees were better informed about the snake than bonobos. First, whereas the first individual to arrive at the snake in each experiment had a similar mean probability to startle in both species (75% in chimpanzees and 70.8% in bonobos), the likelihood to startle for the late arrivers was significantly lower in chimpanzees than in bonobos (interaction species*first to see the snake in model 1a: likelihood ratio test (LRT), $p = 0.008$; table 1). Specifically, late-arriving chimpanzees were three times less likely to startle than late-arriving bonobos (12.9% versus 42.7%; figure 1a). Second, chimpanzee late arrivers received more auditory information about the snake than bonobos as they were more likely to have heard a call before seeing the snake (94% versus 62%, effect of species in model 1b: LRT, $p = 0.012$; table 1; electronic supplementary material, figure S5). However, there was no species difference in the visual information available upon arriving at the snake because the likelihood to have a conspecific on the ground within 5 m of the snake with its body oriented towards the snake (electronic supplementary material. figure S4) when approaching the snake was not

significantly different between chimpanzees and bonobos (55% versus 47%, effect of species in model 1c: LRT, $p = 0.675$; table 1; electronic supplementary material, figure S5).

## (c) Triggers of alarm calls (models 2a and 2b)

Whether another individual already called or not impacted the likelihood to call and the number of calls produced differently in both species (interaction species*someone called: LRT, $p = 0.004$ in model 2a and $p = 0.003$ in model 2b; table 2). Chimpanzees were less likely to call and produced less calls when someone already called whereas bonobos were more likely to call and uttered more calls if someone already called (figure 1b and electronic supplementary material, figure S6). In addition, whereas order in which individuals saw the snake did not have a strong impact on the likelihood to call in chimpanzees, in bonobos, individuals were least likely to call the later they saw the snake in the experiment (interaction species*order of arrival: LRT, $p = 0.027$, model 2a; table 2; electronic supplementary material, figure S7).

Beyond the species differences, individuals of both species were on average three times more likely to call when ignorant individuals were present in the audience (60% in chimpanzees and 65% in bonobos) than when not (27% in chimpanzees and 18% in bonobos; figure 1c) indicated by a trend in model 2a (LRT, $p = 0.066$; table 2) and by a 95% confidence interval of the estimate that did not overlap with 0. Most individuals never called in any situation when there were no ignorant individuals in the audience (11 out of 17, 64.7% of chimpanzees and 10 out of 14,

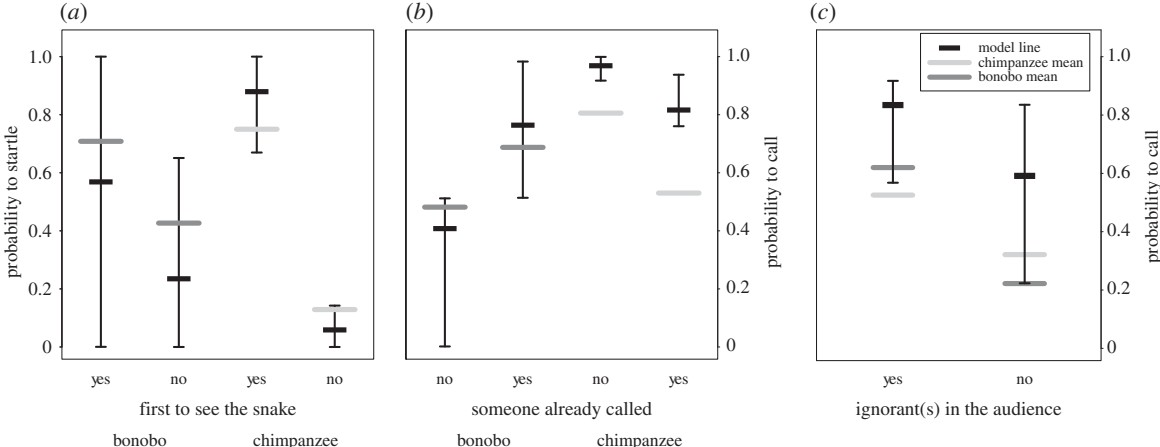

**Figure 1.** Chimpanzees and bonobos startling reaction and calling behaviour upon seeing the snake model: (*a*) probability to startle for the first individual to see the snake (left) and for the late arrivers (right) in both species (model 1a); (*b*) probability to call when someone already called (right) or not (left) in both species (model 2a); and (*c*) probability to call when ignorant in the audience were present (left) or not (right). For all the plots, the thick horizontal black lines depict the model lines (i.e. the probability to startle or to call calculated by the model while controlling for all other variables). The long dark grey and light grey horizontal lines depict the mean from the real data for bonobos and chimpanzees, respectively. The thin vertical lines depict the lower and upper ends of the 95% confidence interval.

**Table 2.** Results of models investigating the triggers of alarm calls (models 2a and 2b). (s.e. indicates the standard error of the estimate for each predictor. The coded level for each categorical predictor is indicated in brackets. Control predictors are italicized. Significant *p*-values (*p* < 0.05) are indicated in bold. Trends (0.5 < *p* < 0.1) are indicated in italics. $CI_{low}$ and $CI_{high}$ indicate the lower and upper limits of the 95% confidence interval for the estimates of each predictor. The 'focal' indicates the potential signaller.)

| model | response | predictor | estimate | s.e. | $CI_{low}$ | $CI_{high}$ | $\chi^2$ | *p* |
|---|---|---|---|---|---|---|---|---|
| 2a | call Y/N | intercept | −7.11 | 1.15 | −10.35 | −5.01 | — | — |
| | | species (chimpanzee) | 5.34 | 1.35 | 5.34 | 2.90 | — | — |
| | | someone already called (yes) | 3.08 | 1.11 | 3.08 | 0.96 | — | — |
| | | order of arrival | −2.01 | 1.03 | −4.64 | 0.06 | — | — |
| | | sex of the focal (male) | 0.14 | 0.48 | −0.82 | 1.26 | 0.08 | 0.780 |
| | | ignorant individual present (yes) | 1.25 | 0.56 | 0.17 | 2.62 | 3.38 | *0.066* |
| | | average number of potential signallers | −0.31 | 0.27 | −1.00 | 0.30 | 1.12 | 0.289 |
| | | species: someone already called | −5.02 | 1.44 | −9.26 | −2.41 | 8.18 | **0.004** |
| | | species: order of arrival | 2.39 | 1.04 | 0.29 | 5.18 | 4.88 | **0.027** |
| | | *age of the focal age (juvenile)* | −1.82 | 0.59 | −3.32 | −0.77 | 9.43 | **0.002** |
| | | *number of snakes seen by the focal* | −0.13 | 0.47 | −1.18 | 0.87 | 0.06 | 0.802 |
| | | *time since the focal has seen the snake* | 0.07 | 0.18 | −6.54 | 0.46 | 0.13 | 0.719 |
| 2b | number of calls uttered | (intercept) | −2.91 | 0.73 | −4.62 | −1.33 | — | — |
| | | species (chimpanzee) | 2.28 | 0.60 | 0.94 | 3.66 | — | — |
| | | someone already called (yes) | 1.50 | 0.43 | 0.48 | 2.56 | — | — |
| | | order of arrival | 0.29 | 0.14 | −0.08 | 0.70 | 2.54 | 0.111 |
| | | sex of the focal (male) | 0.23 | 0.34 | −−0.45 | 0.94 | 0.47 | 0.495 |
| | | ignorant individual present (yes) | −0.48 | 0.55 | −1.70 | 0.74 | 0.61 | 0.435 |
| | | average number of potential signallers | −0.16 | 0.21 | −0.65 | 0.35 | 0.50 | 0.480 |
| | | species: someone called | −2.32 | 0.46 | −3.57 | −1.22 | 9.03 | **0.003** |
| | | *age of the focal (juvenile)* | −0.85 | 0.44 | −1.80 | −0.02 | 3.54 | *0.060* |
| | | *number of snakes seen by the focal* | −0.15 | 0.31 | −0.74 | 0.46 | 0.22 | 0.637 |
| | | *time since the focal has seen the snake* | 0.02 | 0.05 | −8.01 | 0.17 | 0.12 | 0.733 |

71.4% of bonobos). Finally, concerning our control variables, juveniles were less likely to call than adults (LRT, $p = 0.002$, model 2a; table 2). Furthermore, the number of experimental snakes seen before a particular trial and the time since the individuals have seen the snake within a trial did not significantly influence the likelihood to call or the number of calls produced (all $p > 0.6$; table 2).

## 4. Discussion

Our detailed analysis of behavioural reactions of wild bonobos and chimpanzees to a Gaboon viper model revealed some clear differences between the two species, highlighting differing socio-cognitive mechanisms underlying within-group non-collaborative cooperation. Taken together, our results provide empirical support for one of the core concepts of the interdependence hypothesis [2] to species other than humans and therefore indicate that this proposed mechanism for the emergence of cooperation has a deep evolutionary history. Chimpanzees, the more interdependent species, performed better at a cooperative task (here informing conspecifics about a danger) in a non-collaborative context. Specifically, late arriver chimpanzees were less likely to startle and were more likely to have heard an alarm call (i.e. better informed) than late arriver bonobos. Given that the likelihood to startle for the first individual to see the snake was similar in both species, we rule out that this species difference is caused by any species-specific tendency to startle more when seeing snakes.

Instead, the difference in the startling responses of later arrivers possibly stems from a species difference in how individuals adjusted their behavioural responses to the level of information already available to conspecifics. Specifically, chimpanzees transmitted the information about the snake when needed by calling if they had not yet heard someone call, whereas bonobos were less likely to call if they had not yet heard a call.

In our study, we also reduced, to the best of our abilities, the impact of potentially confounding variables. To control for the effect of habitat, we placed the snake model in areas with similar visibility for both species. The lower performance of bonobos at the alarm calling tasks cannot be driven by the lower likelihood to call in juveniles compared to adults in both species (model 2a) because juveniles discovered the snake only in one experiment (5.2%) in bonobos compared to six experiments (27.3%) in chimpanzees. It can also not be explained by gregariousness as party sizes around the snake were smaller in bonobos than in chimpanzees (electronic supplementary material, table S3). Also, in both species, a similar proportion of individuals who heard an alarm call before approaching the snake heard it less than a minute before (81% in chimpanzees and 80% in bonobos), suggesting that the group spread was similar in chimpanzees and bonobos. It is also unlikely that the species differences arose from a higher level of non-vocal social information transfer in chimpanzees. Indeed, upon arriving at the snake the likelihood to have a conspecific oriented towards the snake and within 5 m of the snake was not significantly different in chimpanzees and bonobos, and both chimpanzees and bonobos gestured towards the snake only on one occasion. Finally, better information transfer in chimpanzees is unlikely to arise from a higher propensity of chimpanzees,

as compared to bonobos, to produce alarm calls in general because a recent study found that bonobos were more likely to produce alarm calls upon discovering a novel object than chimpanzees [31].

In our study, we could not control for the presence of bond partners because social relationship data were missing for two communities, but this parameter is unlikely to explain the species difference found in our study. In fact, the presence of bond partners did not increase the likelihood to call in a similar snake experiment on Eastern chimpanzees [26]. For bonobos, we cannot exclude that individuals arriving early at the snake called only if specific group members, such as bond partners or kin, were present around the snake (see below). Yet, more selective production of alarm calls in bonobos as compared to chimpanzees is one of the predictions derived from the interdependence hypothesis.

There are several possible explanations for the species differences found in our study. This could reflect: (i) a lack of ability of bonobos to assess the need of their conspecifics to be informed, (ii) a low intrinsic motivation to cooperate by making every other bonobo aware of the threat, and (iii) a different function for the calls in the two species. The first option is unlikely as both species show abilities to assess what another can see as shown in experiments where chimpanzees and bonobos took advantage of what conspecifics have seen in competitive situations [32,33]. This is confirmed by our findings as individuals of both species tended to be less likely to call when there was no ignorant individual in the audience (i.e. they kept track of who had seen the snake or not). The second option seems more likely, that the lower performance of bonobos in our alarm calling task may reflect a difference in intrinsic motivation to cooperate by informing others (see below). This provides some contrast to the overall view of the interdependence hypothesis by showing that while interdependence does indeed promote more efficient within-group non-collaborative cooperation, this is not necessarily linked to differences in cognitive abilities in two closely related species, at least in terms of attending to other's knowledge state. Instead, interdependence may enhance the motivation to engage in non-collaborative cooperation.

Finally, while a primary motivation to call in both species seems to be to inform ignorant others, we also cannot fully rule out that calls serve additional functions in each species. For example, the fact that in our study, bonobos were more likely to call when someone had already called indicates that they may call also to acknowledge that they have seen the threat (a possible secondary function in Thomas langurs [22] and chimpanzees [25]).

Our behavioural data show that both species tended to be more likely to call when there was an ignorant individual in the audience and is in line with a previous study using the same experimental paradigm but in a different subspecies (i.e. Eastern chimpanzees [26]). We are agnostic as to the underlying socio-cognitive mechanisms required, in particular whether or not this pattern of calling shows that apes can attribute mental states to others (see [29,34]). However, regardless of the precise mechanism, alarm calling patterns in Eastern [26] and Western chimpanzees (our study) and bonobos (our study) suggest a level of attention to the audience perhaps not yet demonstrated in non-ape species.

Non-collaborative cooperative acts (here alarm calling) are expected to be expressed when the benefit for the

recipient outweighs the cost for the cooperator [3]. Clearly, potential death following a bite by a Gaboon viper is costlier than the production of alarm calls. Our results indicate that both species do not differ in the cognitive capacities to call when information is needed by others. Thus, the question is why, unlike chimpanzees, bonobos invest less in calling to make sure that everybody is informed about the threat? The answer probably lies in the difference in the level of between individual interdependence in each species, potentially owing to differences in territoriality. Chimpanzees are highly territorial and are interdependent on each other for coordinated action in territorial activity such as border patrols and to win, potentially lethal, inter-group conflicts. To avoid a large power imbalance that can lead to killing [35], in-group members need to coordinate their actions to reduce potential costs. In chimpanzees [36] and more globally in territorial monkey species and in humans [37], larger groups outcompete smaller neighbouring groups in inter-group encounters. They also have larger territories and benefit from enhanced reproductive success ([36] reviewed in [37]). In Taï chimpanzees, males almost systematically take part in inter-group conflicts and females participate in more than 90% of these events [9]. In this context, the death of an individual of either sex influences the competitive abilities of the community and thereby the fitness of its members. This may increase the incentive for each individual chimpanzee to ensure that every member of the community is informed about the snake in our experiment. By contrast, bonobos do not have a defined territory to defend and inter-group encounters are often peaceful [38]. In this context, the death of a specific individual might not have direct costs for the entire community but may rather affect coalitionary support available to kin (e.g. mothers and sons). Such differences in how fitness gains occur in the two species may explain why, in our snake experiment, bonobos do not attend to the needs of several others simultaneously as systematically as chimpanzees.

In human experimental studies, high levels of between-group competition incentivizes within-group non-collaborative cooperation (e.g. [39]). In our study, bonobos arriving late at the snake were less likely to call whereas the likelihood to call did not decrease for late-arriving chimpanzees (while controlling for if others called or not and the presence of ignorant individuals in the audience). Thus, the responsibility to call rests mostly with those arriving early in bonobos but not in chimpanzees. This result supports the view that individual chimpanzees take more responsibility in making sure everyone is informed about the snake than bonobos. Even if signallers do not coordinate calling per se to achieve a common goal (e.g. calling altogether, which would be seen more as mobbing behaviour towards a threat [27]), in chimpanzees, they collectively succeed at the task to inform everyone, a potential collective action. Thus, the overall success at information transfer depends on the behaviour of each potential signaller. Our results are in line with a meta-analysis on 138 primate species which showed that highly territorial species and those in which the philopatric sex is the dominant (as is the case in our study for the chimpanzee) are more successful at resolving collective action problems [40].

Our results do not support the tolerance hypothesis because bonobos were not more successful than chimpanzees at keeping track of others needs and at informing everyone.

They are also not in line with captive experimental studies showing better abilities of bonobos to collaborate with everyone, including unfamiliar individuals [17], and to succeed in competitive socio-cognitive tasks [14] than chimpanzees. These discrepancies might arise from differences in individual intrinsic motivations to cooperate between wild and captive individuals (see above). Another difference might be linked to the nature of the task and the associated reward. In captive studies, individuals gain direct benefits from completing the task as they usually get access to highly preferred food resources. By contrast, in our experiment, the benefit for the cooperator is delayed substantially with individual pay-offs arising from the overall competitive ability of the community. In captivity, collaboration by paired chimpanzees can be impaired by a lack of tolerance with the dominant individual often monopolizing the reward after collaboration, whereas more tolerant bonobos are more likely to share the reward perhaps facilitating their increased likelihood to engage in collaborative tasks with food rewards [17]. By contrast, in the specific case of our experiment, social tolerance per se may not play a strong role in enhancing non-collaborative cooperative abilities of the bonobos.

The wealth of experimental studies conducted on captive chimpanzees and bonobos (e.g. [14,17,19,41]) shed light on several of the socio-cognitive and cooperative abilities of our two closest relatives and thus have contributed greatly to our understanding of the evolution of unique human cognition and cooperative abilities. Our results bring our knowledge a step further. In fact, we provide for the first, to our knowledge, time empirical support to major theories on the evolution of cooperation in humans using a comparative experiment on wild living individuals from our two closest relatives in an ecologically relevant context. As predicted by the interdependence hypothesis [2], the species which engages more routinely in group-level collaboration (the chimpanzee) performed best at a within-group non-collaborative cooperative task, and this performance was tightly linked to individual behavioural decisions. However, our results indicate that this performance may not stem, as predicted by the interdependence hypothesis, from differences in cognitive capacities between chimpanzees and bonobos but may instead be linked to chimpanzees having a higher motivation to cooperate by informing others of a threat. Finally, the better performance of the more territorial species (the chimpanzee) in an in-group non-collaborative cooperative task highlights the considerable role that inter-group competition may have had on the evolution of the extraordinary in-group cooperative capability observed in humans [42].

**Ethics.** The 'Ethikrat' of the Max Planck Society gave ethical approval on 4 August 2014.

**Data accessibility.** Additional data are provided in the electronic supplementary material. Data are available from the Dryad Digital Repository: https://doi.org/10.5061/dryad.573n5tb43 [43].

**Competing interests.** We declare we have no competing interests.

**Funding.** C.C. and R.M.W. were supported by the European Research Council (ERC) (Grant Agreement No 679787). Core funding for the Taï Chimpanzee Project has been provided by the Max Planck Society since 1997. Core funding for LuiKotale came from the Royal Zoological Society of Antwerp (KMDA) and private donors.

**Acknowledgements.** We thank the Ministère de l'Enseignement Supérieur et de la Recherche Scientifique and the Ministère des Eaux et Forêts in Côte d'Ivoire and the Office Ivoirien des Parcs et Réserves for permitting the study in Taï. We are grateful to the Centre Suisse de Recherches Scientifiques en Côte d'Ivoire. We thank the

Institut Congolais pour la Conservation de la Nature (ICCN) for granting permission to work at LuiKotale and Lompole village for facilitating research in the forest. Many thanks go to Pascalle Dekker, Guilhem Duvot, Adeelia Goffe, Charlotte Grund and Joanna Riera for invaluable assistance in conducting the snake experiments and to Liza Moscovice, Tatiana Thomas, Sean Lee and all the staff members of the Taï Chimpanzee and LuiKotale Bonobo Projects for their help in the field. We are grateful to Marike Schreiber and Lukas Westphal for their help in designing and three-dimensional printing the snake models and to Heather Cohen for blind coding some experimental videos. We thank Julia Fischer and three anonymous reviewers for constructive comments on the manuscript.

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
