## [Reviewer comments · Proceedings of the Royal Society B: Biological Sciences]

Review History

RSPB-2020-0197.R0 (Original submission)

Review form: Reviewer 1

Recommendation

Accept with minor revision (please list in comments)

Scientific importance: Is the manuscript an original and important contribution to its field?

Excellent

General interest: Is the paper of sufficient general interest?

Good

Quality of the paper: Is the overall quality of the paper suitable?

Excellent

Is the length of the paper justified?

Yes

Should the paper be seen by a specialist statistical reviewer?

No

Do you have any concerns about statistical analyses in this paper? If so, please specify them explicitly in your report.

No

It is a condition of publication that authors make their supporting data, code and materials available - either as supplementary material or hosted in an external repository. Please rate, if applicable, the supporting data on the following criteria.

Is it accessible?

Yes

Is it clear?

Yes

Is it adequate?

Yes

Do you have any ethical concerns with this paper?

No

Comments to the Author

Review for Information transfer efficiency differs in wild chimpanzees and bonobos, but not social cognition

General comments

This manuscript uses comparative experimental field research to test two hypotheses (interdependence hypothesis and social tolerance hypothesis) on the origin of cooperative behaviour in apes. The authors compare the performance and socio-cognitive abilities of chimpanzees (more interdependent) and bonobos (more socially tolerant) in a non-collaborative task in the wild, using a snake model paradigm.

I find this an interesting and well-designed study to test two competing hypotheses on the origins of cooperative behaviour in a natural setting. It is rare to see such a study carried out in the wild and on such a sample size in apes. Furthermore, I find the paper well-written and believe it will be of interest to researchers from various domains including animal behaviour and anthropology.

Abstract

L 39 & 40 & 45: You use the terms “collaborate” and “non-collaborative” here. These terms are defined in the main text but not in the abstract, making their use in the abstract confusing for the reader. I think you should either define the terms in the abstract or not use them.

Introduction

Overall the introduction is clearly written and summarises important notions for this study such as the interdependence hypothesis and the social tolerance hypothesis. I do however feel it is a bit long at six pages and in particular the aims section (that extends from line 121 to 188) could be shorter and more concise.

L 136: “danger level to conspecifics”

Methods

L 200: “Democratic Republic of Congo”

L 200: age categories should be defined in the main text

L 224: How many chimpanzees and how many bonobos?

L 224-225 parenthesis: Is this relevant?

L 231-245: The video analysis section should be a single paragraph

L 251: The table referred to should be Table S2

L 277: Remove the extra speech marks

L 293- 296: It would make the four parameters used in this model stand out more for the reader if you added in the numbers 3 and 4 before the last two parameters: “To test if individuals considered who else could inform the audience in their decision to call, we used a third test predictor: 3) the number of other potential signalers. We used 4) “order of arrival” (the first to arrive at the snake on a given experiment day gets a one, the second to arrive a two etc.) to assess if late arrivers were less likely to inform conspecifics than first arrivers.

L 307: I would start a new paragraph here

L 315-316: Add in references for R and the lme4 package

Results

L 342: I would include figure S5 in the main text

L 345: “snake oriented body” is not necessary, it is already noted L 277

L 365: juveniles were less likely to call than what?

Discussion

L 412: “at least in two closely related species”

L 414-418: Could you suggest a way that this explanation could be ruled out?

L 450-451: Do you have the information about the audience composition around the snake and whether the presence of a socially preferred partner influences the likelihood to call in these two species? I would understand if this information is not available and the MS is already rather long, but I think this may give some insight into the difference in alarm calling behaviour between the two species.

L 466: “(as is the case in our study for the chimpanzee)”

Figure 1

The colours used to not translate well to grayscale

The figures 1a and 1b could be made clearer by adding the species labels on the x-axis

There is no y-axis label for figure 1b

As far as I can tell, figure 1b and figure S6 are the same, please remove one of them

Supplemental Material

Fig S5: I would move this figure to the main text

Fig S6: This figure seems to me to be the same as figure 1b in the main text.

Table S2: I find this table very useful to follow the models the authors have used.

Movies S1 and S2: I like the fact that the authors have included these movies in the supplemental material. They give the reader a better idea of the apes’ reactions.

Databases S1, S2 & S3: These databases have multiple empty columns. In order to make it clearer, I would consider deleting these columns as they obviously add no extra information.

Review form: Reviewer 2

Recommendation

Major revision is needed (please make suggestions in comments)

Scientific importance: Is the manuscript an original and important contribution to its field?

Acceptable

General interest: Is the paper of sufficient general interest?

Acceptable

Quality of the paper: Is the overall quality of the paper suitable?

Acceptable

Is the length of the paper justified?

Yes

Should the paper be seen by a specialist statistical reviewer?

No

Do you have any concerns about statistical analyses in this paper? If so, please specify them explicitly in your report.

No

It is a condition of publication that authors make their supporting data, code and materials available - either as supplementary material or hosted in an external repository. Please rate, if applicable, the supporting data on the following criteria.

Is it accessible?

N/A

Is it clear?

N/A

Is it adequate?

N/A

Do you have any ethical concerns with this paper?

No

Comments to the Author

This is an interesting and generally well-written paper. It details experimental manipulations on wild chimpanzee and bonobo populations that aim to evaluate the merits of the interdependence hypothesis and the social tolerance hypothesis. The study found that chimpanzees arriving late at a model snake encounter were less likely to startle, more likely to have heard a call and less likely to produce a call having already heard a call, while the opposite appeared true for bonobos. These were difficult experiments to undertake and the authors are commended for their field setup.

While I enjoyed this paper, I found the results and discussion section to be confusing and at times difficult to follow. Arguments supporting the success of a hypothesis in one species were not always convincingly supported compared to the results found in the other and I was unconvinced by the arguments for/against underlying mechanisms. This may stem from the general confusion and repetition of results relating to alarm calling in subtly different contexts, and does not necessarily negate the value of the study, but does require attention. In its current form, it is therefore difficult to judge the merits of the paper.

It is key for the support of the hypotheses that bonobos at the front of the group/early on in the snake discovery were less likely to call. Whereas bonobos that arrived later were more likely to call and vice versa for chimpanzees. There are many potential explanations/levels to this which were not ruled out. They may be self-explanatory to those familiar with the two species but as a non-primatologist I was left wondering. Had the individuals at the front of the group not seen the snake or did they see it and not produce an alarm call? What was the group order of travel in the two species and might that explain this? E.g. age-related or dominance-related effects, if a juvenile was the first to encounter a threat that could influence their the likelihood of calling. Do bonobos have a higher threshold for calling? Do they show heavier reliance on other senses e.g. visual medium, if they see an individual adopt a physical posture indicating presence of a snake,

are they less likely to call as well?

Line 407-408: This appears difficult to explain alongside the counter-intuitive result that they call more if alarm calls were already heard?

Line 450: Is it possible to know from these experiments whether bonobos were more likely to call if their close kin or coalition partner was uninformed rather than general group members?

Line 458: This appears very similar to the argument used above to explain why bonobos called more than chimpanzees in some instances. "For example, the fact that in our study, bonobos were more likely to call when someone already called indicate that they may (as has been shown in Thomas langurs [23]) call also to acknowledge that they have seen the threat". Could the same not be true for chimpanzees here as well/instead?

Decision letter (RSPB-2020-0197.R0)

27-Feb-2020

Dear Dr Girard-Buttoz:

I am writing to inform you that your manuscript RSPB-2020-0197 entitled "Information transfer efficiency differs in wild chimpanzees and bonobos, but not social-cognition" has, in its current form, been rejected for publication in Proceedings B. This action has been taken on the advice of referees, who have recommended that substantial revisions are necessary. In particular, while I, the reviewers and the AE all find your experiment to be very well done, reviewer 2 raises concerns about the theoretical framing and the degree to which this experiment will be understood by those who are not primatologists. I would be happy to consider a resubmission, provided the comments of the referees are fully addressed.

Sincerely,
Dr Sarah Brosnan
Editor, Proceedings B

Associate Editor

Board Member: 1

Comments to Author:

Both reviewers see merit in the paper, and while R1 thinks that revisions need only be minor, R2 is concerned that the competing hypotheses are not adequately dealt with at present. This is particularly important as it seems that some explanations may rely on readers having an understanding of primate behaviour, which needs to be addressed for the broad audience of this journal.

Reviewer(s)' Comments to Author:

Referee: 1

Comments to the Author(s)

Review for Information transfer efficiency differs in wild chimpanzees and bonobos, but not social cognition

General comments

This manuscript uses comparative experimental field research to test two hypotheses (interdependence hypothesis and social tolerance hypothesis) on the origin of cooperative behaviour in apes. The authors compare the performance and socio-cognitive abilities of chimpanzees (more interdependent) and bonobos (more socially tolerant) in a non-collaborative task in the wild, using a snake model paradigm.

I find this an interesting and well-designed study to test two competing hypotheses on the origins of cooperative behaviour in a natural setting. It is rare to see such a study carried out in the wild and on such a sample size in apes. Furthermore, I find the paper well-written and believe it will be of interest to researchers from various domains including animal behaviour and anthropology.

Abstract

L 39 & 40 & 45: You use the terms “collaborate” and “non-collaborative” here. These terms are defined in the main text but not in the abstract, making their use in the abstract confusing for the reader. I think you should either define the terms in the abstract or not use them.

Introduction

Overall the introduction is clearly written and summarises important notions for this study such as the interdependence hypothesis and the social tolerance hypothesis. I do however feel it is a bit long at six pages and in particular the aims section (that extends from line 121 to 188) could be shorter and more concise.

L 136: “danger level to conspecifics”

Methods

L 200: “Democratic Republic of Congo”

L 200: age categories should be defined in the main text

L 224: How many chimpanzees and how many bonobos?

L 224-225 parenthesis: Is this relevant?

L 231-245: The video analysis section should be a single paragraph

L 251: The table referred to should be Table S2

L 277: Remove the extra speech marks

L 293- 296: It would make the four parameters used in this model stand out more for the reader if you added in the numbers 3 and 4 before the last two parameters: “To test if individuals considered who else could inform the audience in their decision to call, we used a third test

predictor: 3) the number of other potential signalers. We used 4) “order of arrival” (the first to arrive at the snake on a given experiment day gets a one, the second to arrive a two etc.) to assess if late arrivers were less likely to inform conspecifics than first arrivers.

L 307: I would start a new paragraph here

L 315-316: Add in references for R and the lme4 package

Results

L 342: I would include figure S5 in the main text

L 345: “snake oriented body” is not necessary, it is already noted L 277

L 365: juveniles were less likely to call than what?

Discussion

L 412: “at least in two closely related species”

L 414-418: Could you suggest a way that this explanation could be ruled out?

L 450-451: Do you have the information about the audience composition around the snake and whether the presence of a socially preferred partner influences the likelihood to call in these two species? I would understand if this information is not available and the MS is already rather long, but I think this may give some insight into the difference in alarm calling behaviour between the two species.

L 466: “(as is the case in our study for the chimpanzee)”

Figure 1

The colours used to not translate well to grayscale

The figures 1a and 1b could be made clearer by adding the species labels on the x-axis

There is no y-axis label for figure 1b

As far as I can tell, figure 1b and figure S6 are the same, please remove one of them

Supplemental Material

Fig S5: I would move this figure to the main text

Fig S6: This figure seems to me to be the same as figure 1b in the main text.

Table S2: I find this table very useful to follow the models the authors have used.

Movies S1 and S2: I like the fact that the authors have included these movies in the supplemental material. They give the reader a better idea of the apes’ reactions.

Databases S1, S2 & S3: These databases have multiple empty columns. In order to make it clearer, I would consider deleting these columns as they obviously add no extra information.

Referee: 2

Comments to the Author(s)

This is an interesting and generally well-written paper. It details experimental manipulations on wild chimpanzee and bonobo populations that aim to evaluate the merits of the interdependence hypothesis and the social tolerance hypothesis. The study found that chimpanzees arriving late at a model snake encounter were less likely to startle, more likely to have heard a call and less likely to produce a call having already heard a call, while the opposite appeared true for bonobos. These were difficult experiments to undertake and the authors are commended for their field setup.

While I enjoyed this paper, I found the results and discussion section to be confusing and at times difficult to follow. Arguments supporting the success of a hypothesis in one species were not always convincingly supported compared to the results found in the other and I was unconvinced by the arguments for/against underlying mechanisms. This may stem from the general confusion and repetition of results relating to alarm calling in subtly different contexts, and does not necessarily negate the value of the study, but does require attention. In its current form, it is therefore difficult to judge the merits of the paper.

It is key for the support of the hypotheses that bonobos at the front of the group/early on in the snake discovery were less likely to call. Whereas bonobos that arrived later were more likely to call and vice versa for chimpanzees. There are many potential explanations/levels to this which were not ruled out. They may be self-explanatory to those familiar with the two species but as a non-primatologist I was left wondering. Had the individuals at the front of the group not seen the snake or did they see it and not produce an alarm call? What was the group order of travel in the two species and might that explain this? E.g. age-related or dominance-related effects, if a juvenile was the first to encounter a threat that could influence their the likelihood of calling. Do bonobos have a higher threshold for calling? Do they show heavier reliance on other senses e.g visual medium, if they see an individual adopt a physical posture indicating presence of a snake, are they less likely to call as well?

Line 407-408: This appears difficult to explain alongside the counter-intuitive result that they call more if alarm calls were already heard?

Line 450: Is it possible to know from these experiments whether bonobos were more likely to call if their close kin or coalition partner was uninformed rather than general group members?

Line 458: This appears very similar to the argument used above to explain why bonobos called more than chimpanzees in some instances. "For example, the fact that in our study, bonobos were more likely to call when someone already called indicate that they may (as has been shown in Thomas langurs [23]) call also to acknowledge that they have seen the threat". Could the same not be true for chimpanzees here as well/instead?

Author's Response to Decision Letter for (RSPB-2020-0197.R0)

See Appendix A.

RSPB-2020-0523.R0

Review form: Reviewer 3

Recommendation

Major revision is needed (please make suggestions in comments)

Scientific importance: Is the manuscript an original and important contribution to its field?

Excellent

General interest: Is the paper of sufficient general interest?

Excellent

Quality of the paper: Is the overall quality of the paper suitable?

Good

Is the length of the paper justified?

Yes

Should the paper be seen by a specialist statistical reviewer?

No

Do you have any concerns about statistical analyses in this paper? If so, please specify them explicitly in your report.

No

It is a condition of publication that authors make their supporting data, code and materials available - either as supplementary material or hosted in an external repository. Please rate, if applicable, the supporting data on the following criteria.

Is it accessible?

Yes

Is it clear?

Yes

Is it adequate?

Yes

Do you have any ethical concerns with this paper?

No

Comments to the Author

This paper describes comparative experimental field research to test the interdependence hypothesis and social tolerance hypotheses, and therefore get insight into the evolution of cooperation in primates. The authors compare the reaction of wild chimpanzees and bonobos using a snake model paradigm, suggested to represent a non-collaborative cooperative task. The authors found that chimpanzees, the more interdependent species, performed better at the cooperative task (here informing conspecifics about a danger) in this so-called non-collaborative context.

I find that the theoretical framework and the research questions are very interesting and highly relevant for the field of primate cognition research and for a larger audience interested in comparative psychology and animal behaviour. I praise the authors for conducting field studies to investigate cognitive skills in wild non-human primates: this is crucial for our field and still very scarce. The sample size is good (52 chimpanzees and 30 bonobos). The methods and results are clearly described. The models (GLMM) used for the data analysis are, to my knowledge, generally sound. Overall the paper is clear and well written.

While I enjoyed reading the paper, my main concern lies in the interpretation of the findings. I agree that bonobos and chimpanzees represent ideal species to investigate both hypotheses and investigate the evolution of cooperation, however I question the suitability of the task to test these hypotheses. I think the authors should consider more parsimonious explanations for their findings.

Indeed, in my opinion, the context of alarm calling towards a snake model does not represent a non-collaborative cooperative task or a cooperative task at all.

The authors claim that [they] "used a snake model stimulus paradigm allowing the apes to express a non-collaborative cooperative acts, namely to produce alarm calls alerting conspecifics to a deadly threat." (line 123) and further in the text "Since the function of calls produced around vipers is unknown in bonobos, we assumed that the calls also function to inform others, especially given that snake-associated behaviour is broadly comparable to that observed in chimpanzees. Informing individuals present in the vicinity of a danger can be seen as a non-collaborative cooperative task [3]. The success at this task should reflect a species overall cooperative tendency and its socio-cognitive abilities to respond to the needs of others." (line 145)

The authors imply here that the production of an alarm call belongs to a second-order intentionality, i.e. that a caller intentionally alerts a conspecific to a threat. However, this is a highly debatable topic in the literature and to the best of my knowledge, there has been no solid evidence so far for this ability in non-human primates. Actually, most researchers agree that vocalisations in non-human primates are mostly hardwired to particular stimuli and should be at best conceived as goal-directed, where signallers are sensitive to the relation between their signalling and receivers' responses (see for instance a recent review, Fischer J & Price T (2017). Meaning, intention, and inference in primate vocal communication, *Neuroscience & Biobehavioral Reviews* 82: 22-31). The context (or task as defined by the authors) does not require any higher level of socio-cognitive skills, e.g. the attribution of mental state to others or any form of cooperation.

In my opinion, defining the presentation of a snake model as a non-collaborative cooperative task is therefore flawed and the interpretation on socio-cognitive skills, e.g. cooperation here, does not hold. A more parsimonious explanation of the findings could be that the alarm calling pattern of each species might only reflect a difference in general excitement and motivational state of the individuals when facing a potential danger. This would be in line in fact with what the authors state further in the discussion: "Yet, our results indicate that this performance may not stem, as predicted by the interdependence hypothesis, from differences in cognitive capacities between chimpanzees and bonobos but may instead be linked to motivational factors" (line 492)

For this reason, I think the authors should cautiously interpret their findings and amend the paper in general, and the discussion accordingly. In particular, they should revise the text throughout to avoid any attribution of higher socio-cognitive skills when using such a paradigm. Here are for instance some paragraphs that would therefore need revisions:

line 158: "If the interdependence hypothesis holds true, we predict that the more "interdependent" Tai chimpanzees evolved specific cooperation-related cognitive skills (here, keeping track of the behaviour and knowledge status of all present group members). Accordingly, chimpanzees are expected to be better at solving the alarm calling task (i.e. informing all other conspecifics)."

line 273: "In Model 2a we assessed which socio-cognitive parameters affected the general decision to call or not for each individual during the experiment."

line 405: "Therefore, the second option seems more likely, that the lower performance of bonobos in our alarm calling task may reflect a difference in intrinsic motivation of individuals to inform everyone (see below) rather than differences in cognitive abilities to track who needs to be informed."

line 415: "Finally, whilst a primary motivation to call in both species seems to be to inform ignorant others"

line 430: "Non-collaborative cooperative acts (here alarm calling) are expected to be expressed when the benefit for the recipient outweighs the cost for the cooperator [3]. Clearly, potential death following a bite by a Gaboon viper is costlier than the production of alarm calls. Our results indicate that both species appear to have the cognitive capacities to assess the knowledge status of conspecifics."

Other minor corrections:

Line 188: '.' Missing after "age categories for both species".

Review form: Reviewer 4

Recommendation

Accept as is

Scientific importance: Is the manuscript an original and important contribution to its field?

Good

General interest: Is the paper of sufficient general interest?

Excellent

Quality of the paper: Is the overall quality of the paper suitable?

Excellent

Is the length of the paper justified?

Yes

Should the paper be seen by a specialist statistical reviewer?

No

Do you have any concerns about statistical analyses in this paper? If so, please specify them explicitly in your report.

No

It is a condition of publication that authors make their supporting data, code and materials available - either as supplementary material or hosted in an external repository. Please rate, if applicable, the supporting data on the following criteria.

Is it accessible?

Yes

Is it clear?

Yes

Is it adequate?

Yes

Do you have any ethical concerns with this paper?

No

Comments to the Author

The authors have done an excellent job when revising the manuscript. I have no further questions or objections.

Decision letter (RSPB-2020-0523.R0)

24-Apr-2020

Dear Dr Girard-Buttoz:

Your manuscript has now been peer reviewed and the reviews have been assessed by an Associate Editor. The reviewers' comments (not including confidential comments to the Editor) and the comments from the Associate Editor are included at the end of this email for your

reference. We have gotten two reviews on your revised manuscript, and as you will see, they are contradictory. The associate editor and I have discussed this, and feel that while overall we really like your study and the potential it offers, that there are other perspectives and interpretations that need to be more fully addressed. Although you should feel free, of course, to maintain your current conclusions, your paper will be far stronger if you fully engage with the points that reviewer 1 brings up, as certainly this individual is not the only person who will have such concerns. Indeed, given the general audience that reads Proceedings, it will be important to engage with those outside of primatology, who might interpret similar findings differently in their own species. Therefore, we are asking you to revise your manuscript to thoroughly consider these alternate perspectives and explain why you reach the conclusion that you do.

Research ethics:

Use of animals and field studies:

Please submit a copy of your revised paper within three weeks. If we do not hear from you within this time your manuscript will be rejected. If you are unable to meet this deadline please let us know as soon as possible, as we may be able to grant a short extension.

Best wishes,
Dr Sarah Brosnan
Editor, Proceedings B
mailto: proceedingsb@royalsociety.org

Associate Editor Board Member: 1
Comments to Author:

Both reviewers see substantial merit in the paper, but R1 has strong reservations about the extent to which the data meet the conclusions. Therefore, in my view, the conclusions need to be much more cautious and alternative explanations considered in more detail before publication.

Reviewer(s)' Comments to Author:

Referee: 3

Comments to the Author(s).

This paper describes comparative experimental field research to test the interdependence hypothesis and social tolerance hypotheses, and therefore get insight into the evolution of cooperation in primates. The authors compare the reaction of wild chimpanzees and bonobos using a snake model paradigm, suggested to represent a non-collaborative cooperative task. The authors found that chimpanzees, the more interdependent species, performed better at the

cooperative task (here informing conspecifics about a danger) in this so-called non-collaborative context.

I find that the theoretical framework and the research questions are very interesting and highly relevant for the field of primate cognition research and for a larger audience interested in comparative psychology and animal behaviour. I praise the authors for conducting field studies to investigate cognitive skills in wild non-human primates: this is crucial for our field and still very scarce. The sample size is good (52 chimpanzees and 30 bonobos). The methods and results are clearly described. The models (GLMM) used for the data analysis are, to my knowledge, generally sound. Overall the paper is clear and well written.

While I enjoyed reading the paper, my main concern lies in the interpretation of the findings. I agree that bonobos and chimpanzees represent ideal species to investigate both hypotheses and investigate the evolution of cooperation, however I question the suitability of the task to test these hypotheses. I think the authors should consider more parsimonious explanations for their findings.

Indeed, in my opinion, the context of alarm calling towards a snake model does not represent a non-collaborative cooperative task or a cooperative task at all.

The authors claim that [they] “used a snake model stimulus paradigm allowing the apes to express a non-collaborative cooperative acts, namely to produce alarm calls alerting conspecifics to a deadly threat.” (line 123) and further in the text “Since the function of calls produced around vipers is unknown in bonobos, we assumed that the calls also function to inform others, especially given that snake-associated behaviour is broadly comparable to that observed in chimpanzees. Informing individuals present in the vicinity of a danger can be seen as a non-collaborative cooperative task [3]. The success at this task should reflect a species overall cooperative tendency and its socio-cognitive abilities to respond to the needs of others.” (line 145) The authors imply here that the production of an alarm call belongs to a second-order intentionality, i.e. that a caller intentionally alerts a conspecific to a threat. However, this is a highly debatable topic in the literature and to the best of my knowledge, there has been no solid evidence so far for this ability in non-human primates. Actually, most researchers agree that vocalisations in non-human primates are mostly hardwired to particular stimuli and should be at best conceived as goal-directed, where signallers are sensitive to the relation between their signalling and receivers’ responses (see for instance a recent review, Fischer J & Price T (2017). Meaning, intention, and inference in primate vocal communication, *Neuroscience & Biobehavioral Reviews* 82: 22-31). The context (or task as defined by the authors) does not require any higher level of socio-cognitive skills, e.g. the attribution of mental state to others or any form of cooperation.

In my opinion, defining the presentation of a snake model as a non-collaborative cooperative task is therefore flawed and the interpretation on socio-cognitive skills, e.g. cooperation here, does not hold. A more parsimonious explanation of the findings could be that the alarm calling pattern of each species might only reflect a difference in general excitement and motivational state of the individuals when facing a potential danger. This would be in line in fact with what the authors state further in the discussion: “Yet, our results indicate that this performance may not stem, as predicted by the interdependence hypothesis, from differences in cognitive capacities between chimpanzees and bonobos but may instead be linked to motivational factors” (line 492)

For this reason, I think the authors should cautiously interpret their findings and amend the paper in general, and the discussion accordingly. In particular, they should revise the text throughout to avoid any attribution of higher socio-cognitive skills when using such a paradigm. Here are for instance some paragraphs that would therefore need revisions:

line 158: “If the interdependence hypothesis holds true, we predict that the more “interdependent” Taï chimpanzees evolved specific cooperation-related cognitive skills (here, keeping track of the behaviour and knowledge status of all present group members).

Accordingly, chimpanzees are expected to be better at solving the alarm calling task (i.e. informing all other conspecifics)."

line 273: "In Model 2a we assessed which socio-cognitive parameters affected the general decision to call or not for each individual during the experiment."

line 405: "Therefore, the second option seems more likely, that the lower performance of bonobos in our alarm calling task may reflect a difference in intrinsic motivation of individuals to inform everyone (see below) rather than differences in cognitive abilities to track who needs to be informed."

line 415: "Finally, whilst a primary motivation to call in both species seems to be to inform ignorant others"

line 430: "Non-collaborative cooperative acts (here alarm calling) are expected to be expressed when the benefit for the recipient outweighs the cost for the cooperator [3]. Clearly, potential death following a bite by a Gaboon viper is costlier than the production of alarm calls. Our results indicate that both species appear to have the cognitive capacities to assess the knowledge status of conspecifics."

Other minor corrections:

Line 188: '.' Missing after "age categories for both species".

Referee: 4

Comments to the Author(s).

The authors have done an excellent job when revising the manuscript. I have no further questions or objections.

Author's Response to Decision Letter for (RSPB-2020-0523.R0)

See Appendix B.

Decision letter (RSPB-2020-0523.R1)

24-May-2020

Dear Dr Girard-Buttoz

I am pleased to inform you that your manuscript entitled "Information transfer efficiency differs in wild chimpanzees and bonobos, but not social-cognition" has been accepted for publication in Proceedings B. I appreciate the effort you put into your revision and look forward to seeing your paper in press.

Open Access

Paper charges

Sincerely,

Dr Sarah Brosnan

Associate Editor:

Board Member

Comments to Author:

The concerns of the reviewers have been addressed well and the conclusions are now more conservative and cautious, which is appropriate in light of potential different interpretations.

Appendix A

Both reviewers see merit in the paper, and while R1 thinks that revisions need only be minor, R2 is concerned that the competing hypotheses are not adequately dealt with at present. This is particularly important as it seems that some explanations may rely on readers having an understanding of primate behaviour, which needs to be addressed for the broad audience of this journal.

Reviewer(s)' Comments to Author:

Referee: 1

Comments to the Author(s)

Review for Information transfer efficiency differs in wild chimpanzees and bonobos, but not social cognition

General comments

This manuscript uses comparative experimental field research to test two hypotheses (interdependence hypothesis and social tolerance hypothesis) on the origin of cooperative behaviour in apes. The authors compare the performance and socio-cognitive abilities of chimpanzees (more interdependent) and bonobos (more socially tolerant) in a non-collaborative task in the wild, using a snake model paradigm. I find this an interesting and well-designed study to test two competing hypotheses on the origins of cooperative behaviour in a natural setting. It is rare to see such a study carried out in the wild and on such a sample size in apes. Furthermore, I find the paper well-written and believe it will be of interest to researchers from various domains including animal behaviour and anthropology.

We thank the reviewer for the positive evaluation of our manuscript.

Abstract

L 39 & 40 & 45: You use the terms “collaborate” and “non-collaborative” here. These terms are defined in the main text but not in the abstract, making their use in the abstract confusing for the reader. I think you should either define the terms in the abstract or not use them.

We thank the reviewer for this suggestion. We have removed the term collaborative and non-collaborative from the abstract.

Introduction

Overall the introduction is clearly written and summarises important notions for this study such as the interdependence hypothesis and the social tolerance hypothesis. I do however feel it is a bit long at six pages and in particular the aims section (that extends from line 121 to 188) could be shorter and more concise.

As suggested by the reviewer we have now shortened our introduction.

L 136: "danger level to conspecifics"

This has been changed (line 130).

Methods

L 200: "Democratic Republic of Congo"

This has been changed (Line 187).

L 200: age categories should be defined in the main text

We have added descriptions of age categories to the main text (Lines 187-192).

L 224: How many chimpanzees and how many bonobos?

We have added this information to the manuscript (lines 214-215).

L 224-225 parenthesis: Is this relevant?

We agree with the reviewer that this information might be irrelevant; we removed it from the manuscript.

L 231-245: The video analysis section should be a single paragraph

We made a single paragraph out of the video analysis section (Lines 221-234).

L 251: The table referred to should be Table S2

We thank the reviewer for his/her attention to details. We have changed the referred table to table S2 (Line 240).

L 277: Remove the extra speech marks

We have removed the extra speech marks (Lines 265-266).

L 293- 296: It would make the four parameters used in this model stand out more for the reader if you added in the numbers 3 and 4 before the last two parameters: "To test if individuals considered who else

could inform the audience in their decision to call, we used a third test predictor: 3) the number of other potential signalers. We used 4) “order of arrival” (the first to arrive at the snake on a given experiment day gets a one, the second to arrive a two etc.) to assess if late arrivers were less likely to inform conspecifics than first arrivers.

This has been changed according to the reviewer’s suggestion (Lines 279-288).

L 307: I would start a new paragraph here

We started a new paragraph where suggested (Line 298).

L 315-316: Add in references for R and the lme4 package

We added the appropriate references (Lines 306-307).

Results

L 342: I would include figure S5 in the main text

We agree with the reviewer that it would be nice to include the Figure S5 in the main text. Nevertheless we feel that our study is relatively complex in nature and needs some relatively extended explanation. Therefore we do not think that we have the space to incorporate and extra Figure in the main text of the manuscript.

L 345: “snake oriented body” is not necessary, it is already noted L 277

We removed “snake oriented body” as suggested by the reviewer.

L 365: juveniles were less likely to call than what?

We clarified that juveniles are less likely to call than adults (Line 350).

Discussion

L 412: “at least in two closely related species”

We changed our sentence according to the reviewer’s suggestion (Lines 411-412).

L 414-418: Could you suggest a way that this explanation could be ruled out?

In both species apes are more likely to call when there are ignorant individual present in the audience and most of them stop calling when all conspecifics have seen the snake. This indicates that the primary function of alarm call is to inform others in both species. Yet we have now clarified that we cannot rule out that some call may also be used to acknowledge that the individual has seen the threat in both species (Lines 415-419).

L 450-451: Do you have the information about the audience composition around the snake and whether the presence of a socially preferred partner influences the likelihood to call in these two species? I would understand if this information is not available and the MS is already rather long, but I think this may give some insight into the difference in alarm calling behaviour between the two species.

We did not have information on social relationship strength in all dyads from our study communities since we recorded focal data only on adult individuals and only on one bonobo and two chimpanzee communities. When we subset our dataset only to individuals for whom we have recorded focal behavioral data we are left only with 124 data points (out of 233 originally) which is not enough given the complexity of our model (13 predictors) to provide a reliable estimate of the effect. Nevertheless, we have now run an additional model on this 124 data points with the same structure as model 2a in the paper but adding the predictor “social bond strength” which is the highest dyadic social bond score from the focal perspective with any adult present in the party around the snake. This model is not fully statistically sound since we have 14 predictors for 124 data points but it indicates that it is highly unlikely that social bond strength had a strong effect in our dataset since it was clearly not significant in this model ($P=0.38$). Most importantly, the effect of “someone already called * Species” remained significant ($P=0.049$) despite the small sample size. Since this model is statistically “borderline” and may not provide fully meaningful information about our data we would like to keep this model out of the current manuscript.

However, we understand the importance of discussing the potential effect of social bonds and we have now added a paragraph about this in the discussion (Line 390-397).

We think that the species differences in the social triggers of calling behavior (i.e. that chimpanzees are more likely to call when no one called before them and that bonobos call more when someone called before) is unlikely to be related to the presence of bond partners.

First, in another snake experiment conducted on another population of chimpanzees [1], the presence of bond partner influenced only the number of calls produced (for a subset of individuals who did call) but not the likelihood to call or not. The information transfer efficiency (i.e. if individual heard an alarm call before seeing the snake) is likely to be affected more by the fact that others called or not than by the number of calls produced by the one who called. Therefore, the presence of bond partners is unlikely to affect the core result of our study.

Second, the average likelihood for a given individual to call or not during a given experiment was virtually identical between the two species (41.7% in the chimpanzees and 40% in the bonobos). The main difference is, therefore, in when they call in the experiment and, in particular, the individuals arriving at the snake early in bonobos call less often than the one arriving early in the chimpanzees. In fact, individuals arriving second or later at the snake were less likely to have heard an alarm call in bonobos as compared to chimpanzees (Model 1b). The species difference is therefore driven by how individuals adjust their behavior to the behavior of others and not by the overall likelihood of each individual to produce alarm call. Even if the presence of bond partners would increase the likelihood for individuals to call, that would not explain the species difference we found here.

Third, social bond and cooperation patterns are associated differently in bonobos and chimpanzees which make it difficult to interpret comparatively the potential effect of social bond in our cooperative experiment. While chimpanzees bond partners often engage into coalitions [2], social bond strength does not predict coalition formation in bonobos [3].

L 466: “(as is the case in our study for the chimpanzee)”

This has been changed according to the reviewer’s suggestion (Line 466).

Figure 1

The colours used to not translate well to grayscale

Thank you for pointing that out. We have changed the color in Figure 1 to grades of grey and black. We kept the original colors in the supplementary material figures since they will only appear online.

The figures 1a and 1b could be made clearer by adding the species labels on the x-axis

We added the species name on the x-axis of Figures 1a and 1b.

There is no y-axis label for figure 1b

We added a label on the y-axis of figure 1b.

As far as I can tell, figure 1b and figure S6 are the same, please remove one of them

Figure 1b and S6 are different. While they both refer to the effect of someone already calling or not in both species, the response variables (i.e. the Y axis in the Figures) differ and each figure represents a different model. Figure 1b depicts the likelihood to call or not (Model 2a) while Figure S6 depicts the number of calls produced (Model 2b).

Supplemental Material

Fig S5: I would move this figure to the main text

Due to space limitation we would like to keep this figure in the supplementary material (see above)

Fig S6: This figure seems to me to be the same as figure 1b in the main text.

This figure is different from figure 1b (See above).

Table S2: I find this table very useful to follow the models the authors have used.

We thank the reviewer for the positive feedback.

Movies S1 and S2: I like the fact that the authors have included these movies in the supplemental material. They give the reader a better idea of the apes' reactions.

We are grateful to the reviewer for providing encouraging feedback.

Databases S1, S2 & S3: These databases have multiple empty columns. In order to make it clearer, I would consider deleting these columns as they obviously add no extra information.

We thank the reviewer for pointing that out. We have now deleted the empty columns in the databases.

Referee: 2

Comments to the Author(s)

This is an interesting and generally well-written paper. It details experimental manipulations on wild chimpanzee and bonobo populations that aim to evaluate the merits of the interdependence hypothesis

and the social tolerance hypothesis. The study found that chimpanzees arriving late at a model snake encounter were less likely to startle, more likely to have heard a call and less likely to produce a call having already heard a call, while the opposite appeared true for bonobos. These were difficult experiments to undertake and the authors are commended for their field setup.

We thank the reviewer for the positive evaluation of our manuscript.

While I enjoyed this paper, I found the results and discussion section to be confusing and at times difficult to follow. Arguments supporting the success of a hypothesis in one species were not always convincingly supported compared to the results found in the other and I was unconvinced by the arguments for/against underlying mechanisms. This may stem from the general confusion and repetition of results relating to alarm calling in subtly different contexts, and does not necessarily negate the value of the study, but does require attention. In its current form, it is therefore difficult to judge the merits of the paper.

We have clarified the results section and rephrased part of the discussion to better link our results to the hypothesis tested.

It is key for the support of the hypotheses that bonobos at the front of the group/early on in the snake discovery were less likely to call. Whereas bonobos that arrived later were more likely to call and vice versa for chimpanzees. There are many potential explanations/levels to this which were not ruled out. They may be self-explanatory to those familiar with the two species but as a non-primatologist I was left wondering.

We apologize for the confusion and we have revised the manuscript to clarify the implications of our results in relation to our hypotheses (see below).

Had the individuals at the front of the group not seen the snake or did they see it and not produce an alarm call?

In our analysis we were interested only in the behavior of individuals who saw the snake not those who approach the area around the snake but did not see it. We understand that the variable “order of arrival at the snake” was not explained clearly enough. We have clarified that this relates only to individuals who saw the snake in the manuscript (Lines 249, 285-288, 341, 342).

What was the group order of travel in the two species and might that explain this? E.g. age-related or dominance-related effects, if a juvenile was the first to encounter a threat that could influence their the likelihood of calling.

In our two models investigating the factors influencing the likelihood to call in the two species (Model 2a and 2b), we controlled for order of arrival (i.e. the order at which individuals see the snake) and for

the age and sex of the individual. In these models, the fact that bonobos are more likely to call when someone already called and that chimpanzees are less likely to call when someone already called, can thus not be explained by the sex or the age of the individual since both are controlled for in the model.

However, we understand the critic of the reviewer that if e.g. juveniles are systematically the first discoverer of the snake in bonobos but not in chimpanzees, and since juveniles are in general less likely to call in our study (Model 2a), this could explain why individuals arriving late at the snake are less likely to be informed in bonobos than in chimpanzees.

We have investigated this possibility now. In both species, adults discover the snake in most experiments but the percentage of experiments during which juvenile discovered the snake is lower in bonobos compared to chimpanzees (1 experiment i.e. 5.2% in bonobos and 6 experiments, i.e. 27.3 % in chimpanzees). If the age of the first discoverer of the snake would drive the likelihood for conspecifics to be informed, we would expect chimpanzees to be less well informed than bonobos (since juvenile are less likely to call in both species in our study, Model 2a). Nevertheless we found the opposite effect with chimpanzees being more likely to have heard an alarm call before seeing the snake than bonobos. We now added this element to our discussion (Lines 369-373).

Furthermore, it is important to note that apes can be informed not only by the first individual who discovers the snake but also by any individual who saw the snake before them. For instance, if an ape arrives at the snake 5th, he could have been informed by four other individuals, the first who saw the snake but also the second the third and the fourth. This highlights that the behavior of the first individual to see the snake alone cannot explain the difference in information transfer efficiency between the bonobos and the chimpanzees.

Do bonobos have a higher threshold for calling? Do they show heavier reliance on other senses e.g visual medium, if they see an individual adopt a physical posture indicating presence of a snake, are they less likely to call as well?

A recent study investigating the reaction of chimpanzees and bonobos to camera trap in natural context suggests that bonobos may actually have a lower threshold for producing alarm calls than chimpanzees. This study is now mentioned in the discussion.

Line 407-408: This appears difficult to explain alongside the counter-intuitive result that they call more if alarm calls were already heard?

This result may sound counterintuitive but please bear in mind that in our statistical model the effect of “order of arrival at the snake” and of “someone already called yes or no” are controlled for each other. That means that, the effect of order of arrival is evaluated while adjusting for the fact that someone called already or not. This also means that the effect of “someone called yes or no” is evaluated within each given order of arrival (e.g. the second to arrive). For instance, a bonobo arriving

second at the snake who heard an alarm call before will have more chance to call than a bonobo arriving fourth at the snake who heard an alarm call. It also means that a bonobo arriving second at the snake who did not hear an alarm call before will have more chance to call than a bonobo arriving fourth at the snake who heard an alarm call. In turn, this does not mean that the second arrivers are, in general, more likely to call than fourth arriver but, after controlling for the effect of hearing an alarm call, they are. In sum, the two effects are independent of each other and controlling for each other. We have added this notion to the discussion (Line 457-458).

Line 450: Is it possible to know from these experiments whether bonobos were more likely to call if their close kin or coalition partner was uninformed rather than general group members?

Unfortunately, we cannot assess the effect of social relationships (or coalitionary partner) on the likelihood to call since we did not record focal behavioral data in all the study communities and our data collection focused on adult individuals. Nevertheless, we do not think that this is explaining the results of our study (see our more detailed reply to reviewer's 1 comments). We have also added a paragraph in the discussion of our manuscript to clarify this specific point (Lines 390-397).

Line 458: This appears very similar to the argument used above to explain why bonobos called more than chimpanzees in some instances. "For example, the fact that in our study, bonobos were more likely to call when someone already called indicate that they may (as has been shown in Thomas langurs [23]) call also to acknowledge that they have seen the threat". Could the same not be true for chimpanzees here as well/instead?

Given the effect of audience knowledge on calling behavior in chimpanzees and bonobos, we think that our results support the idea that most calls are produced to inform others in both species (see also reply to reviewer 1 comments). Nevertheless, we agree with the reviewer that chimpanzees, as bonobos, may produce some calls to acknowledge that they have seen the snake (especially late arriving chimpanzees). We have now added this possibility in our discussion (Line 416-419).

Bibliography

1. Crockford C, Wittig RM, Mundry R, Zuberbuehler K. 2012 Wild chimpanzees inform ignorant group members of danger. *Curr. Biol.* **22**, 142–146. (doi:10.1016/j.cub.2011.11.053)
2. Mitani JC. 2009 Male chimpanzees form enduring and equitable social bonds. *Anim. Behav.* **77**, 633–640. (doi:10.1016/j.anbehav.2008.11.021)

3. Moscovice LR, Douglas PH, Martinez-Iñigo L, Surbeck M, Vigilant L, Hohmann G. 2017 Stable and fluctuating social preferences and implications for cooperation among female bonobos at LuiKotale, Salonga National Park, DRC. *Am. J. Phys. Anthropol.* **163**, 158–172. (doi:10.1002/ajpa.23197)

Appendix B

Please find below the editor's and reviewers' comments in italic and the replies to the comments in bold:

Dear Dr Girard-Buttoz:

Your manuscript has now been peer reviewed and the reviews have been assessed by an Associate Editor. The reviewers' comments (not including confidential comments to the Editor) and the comments from the Associate Editor are included at the end of this email for your reference. We have gotten two reviews on your revised manuscript, and as you will see, they are contradictory. The associate editor and I have discussed this, and feel that while overall we really like your study and the potential it offers, that there are other perspectives and interpretations that need to be more fully addressed. Although you should feel free, of course, to maintain your current conclusions, your paper will be far stronger if you fully engage with the points that reviewer 1 brings up, as certainly this individual is not the only person who will have such concerns. Indeed, given the general audience that reads Proceedings, it will be important to engage with those outside of primatology, who might interpret similar findings differently in their own species. Therefore, we are asking you to revise your manuscript to thoroughly consider these alternate perspectives and explain why you reach the conclusion that you do.

We thank the editors for the positive evaluation of our manuscript. We agree that the concerns raised by the reviewer needed to be addressed. Accordingly, we have amended the manuscript by rephrasing the ambiguous paragraphs highlighted by the reviewer. We have clarified that we did not intend to imply that our study demonstrates mental state attribution in chimpanzees and bonobos. We also specified that we do not refer to arousal when using the term "motivation" but to the eagerness of an individual to produce an alarm call.

Research ethics:

Use of animals and field studies:

If you wish to submit your data to Dryad (<http://datadryad.org/>) and have not already done so you can submit your data via this link [http://datadryad.org/submit?journalID=RSPB&manu=\(Document](http://datadryad.org/submit?journalID=RSPB&manu=(Document) not available), which will take you to your unique entry in the Dryad repository.

Please submit a copy of your revised paper within three weeks. If we do not hear from you within this time your manuscript will be rejected. If you are unable to meet this deadline please let us know as soon as possible, as we may be able to grant a short extension.

Best wishes,

*Dr Sarah Brosnan
Editor, Proceedings B
mailto: proceedingsb@royalsociety.org*

Associate Editor Board Member: 1

Comments to Author:

Both reviewers see substantial merit in the paper, but R1 has strong reservations about the extent to which the data meet the conclusions. Therefore, in my view, the conclusions need to be much more cautious and alternative explanations considered in more detail before publication.

Reviewer(s)' Comments to Author:

Referee: 3

Comments to the Author(s).

This paper describes comparative experimental field research to test the interdependence hypothesis and social tolerance hypotheses, and therefore get insight into the evolution of cooperation in primates. The authors compare the reaction of wild chimpanzees and bonobos using a snake model paradigm, suggested to represent a non-collaborative cooperative task. The authors found that chimpanzees, the more interdependent species, performed better at the cooperative task (here informing conspecifics about a danger) in this so-called non-collaborative context.

I find that the theoretical framework and the research questions are very interesting and highly relevant for the field of primate cognition research and for a larger audience interested in comparative psychology and animal behaviour. I praise the authors for conducting field studies to investigate cognitive skills in wild non-human primates: this is crucial for our field and still very scarce. The sample size is good (52 chimpanzees and 30 bonobos). The methods and results are clearly described. The models (GLMM) used for the data analysis are, to my knowledge, generally sound. Overall the paper is clear and well written.

We thank the reviewer for this overall positive evaluation of the manuscript.

While I enjoyed reading the paper, my main concern lies in the interpretation of the findings. I agree that bonobos and chimpanzees represent ideal species to investigate both hypotheses and investigate the evolution of cooperation, however I question the suitability of the task to test these hypotheses. I think the authors should consider more parsimonious explanations for their findings.

Indeed, in my opinion, the context of alarm calling towards a snake model does not represent a non-collaborative cooperative task or a cooperative task at all.

We apologize for the ambiguity that has arisen here. We use a definition of cooperation common in behavioural ecology studies, which remains agnostic as to psychological processes involved. This is the term “non-collaborative cooperation” which refers to “a service provided by an individual (the cooperator) to a conspecific and which provides some form of fitness benefit to the recipient” (Lines 70-71). When individuals produce alarm calls they inform others about the presence of a danger thereby reducing the likelihood for the conspecifics to be bitten by the snake. In this sense, the individual producing an alarm call is a cooperator which provides a service to the recipient in the form of alarm calls whereby the recipient directly benefits from the service. Alarm calling is broadly used as an example of cooperative behavior in vertebrates (e.g. [1,2]) and Darwin considered that “the most common service which the higher animals perform for each other, is the warning each other of danger by means of the united sense of all” (page 74, [3]). Alarm calling is broadly used across birds, large and small mammals, and does not require any mental state attribution but can operate on much simpler processes. In their elaboration of the interdependence hypothesis, Tomasello and colleagues provide as an example that the likelihood to alarm call should be linked to the level of interdependence since it is in an individual’s own interest to keep potential future collaborators alive [4].

The authors claim that [they] “used a snake model stimulus paradigm allowing the apes to express a non-collaborative cooperative acts, namely to produce alarm calls alerting conspecifics to a deadly threat.” (line 123) and further in the text “Since the function of calls produced around vipers is unknown in bonobos, we assumed that the calls also function to inform others, especially given that snake-associated behaviour is broadly comparable to that observed in chimpanzees. Informing individuals present in the vicinity of a danger can be seen as a non-collaborative cooperative task [3]. The success at this task should reflect a species overall cooperative tendency and its socio-cognitive abilities to respond to the needs of others.” (line 145)

The authors imply here that the production of an alarm call belongs to a second-order intentionality, i.e. that a caller intentionally alerts a conspecific to a threat.

*However, this is a highly debatable topic in the literature and to the best of my knowledge, there has been no solid evidence so far for this ability in non-human primates. Actually, most researchers agree that vocalisations in non-human primates are mostly hardwired to particular stimuli and should be at best conceived as goal-directed, where signallers are sensitive to the relation between their signalling and receivers’ responses (see for instance a recent review, Fischer J & Price T (2017). Meaning, intention, and inference in primate vocal communication, *Neuroscience & Biobehavioral Reviews* 82: 22-31). The*

context (or task as defined by the authors) does not require any higher level of socio-cognitive skills, e.g. the attribution of mental state to others or any form of cooperation.

We are in full agreement with the reviewer that the question of whether or not any species besides humans has second-order intentionality remains a hotly debated topic. Whether or not second-order intentionality exists in apes requires substantial additional testing and particularly in paradigms of false belief. This current manuscript does not address false beliefs and does not attempt to test second-order intentionality in the realm of attribution of knowledge. To avoid further confusion on this point, we have now explicitly stated this in the introduction and in the discussion (Lines 153-154 and 425-427).

As mentioned in the manuscript: “Birds [5] and mammals [6,7] produce alarm calls only in the presence of others” (Lines 127.128). This highlights that alarm calls primarily functions to inform conspecifics about a danger but does not require the caller to necessarily have a mental representation of the audience needs and does not imply second order intentionality. We have rephrased the sentence mentioned by the reviewer to clarify this ambiguity by specifically stated that “We are not aiming to investigate in our study the detail of what specific socio-cognition is involved in completing the task” (Lines 151-154).

Yet result from previous studies on chimpanzees – and now in addition with our study on bonobos – showing that they are more likely to emit alert calls when others have not yet seen the snake, does suggest an additional level of attention is being paid to receivers than has not previously been demonstrated in other species in alarm calling contexts. We are careful to state in the discussion that whilst this appears to be the case, and indicates some complex socio-cognitive processes, we cannot specify from our data, which assesses behaviour and not cognition, what those might be (Lines 425-427). Given that the data suggest a level of careful attention to the audience, which has not yet been demonstrated in non-ape species, we feel that these findings are a valid and interesting addition to the literature. How this level of attention is achieved on a cognitive level remains open for debate and this has been mentioned in the discussion (Lines 425-427).

We have also mentioned in the discussion the paper by Fisher and Price suggested by the reviewer in this section (Line 427) to elaborate on the limitation of our results. However, as explained above, the production of alarm calls is a cooperative behaviour. We also think that, regardless of the mechanisms, our study clearly indicates an ability for the ape to perceive, to some extent, the audience knowledge about the snake and to adjust their alarm calling behaviour accordingly. This assessment does not require mental state attribution but can still be seen as a socio-cognitive ability.

In my opinion, defining the presentation of a snake model as a non-collaborative cooperative task is therefore flawed and the interpretation on socio-cognitive skills, e.g. cooperation here, does not hold. A more parsimonious explanation of the findings could be that the alarm calling pattern of each species might only reflect a difference in general excitement and motivational state of the individuals when facing a potential danger. This would be in line in fact with what the authors state further in the discussion: “Yet, our results indicate that this performance may not stem, as predicted by the interdependence hypothesis, from differences in cognitive capacities between chimpanzees and bonobos but may instead be linked to motivational factors” (line 492).

Again, apologies for the confusion caused by our writing. Our main aim was to refer to “the motivation to cooperate” (again perhaps more a behavioural ecological concept), rather than arousal-related interpretation of “motivation”. We have now clarified this throughout the manuscript (Lines 404, 411, 416, 418, 475, 498). Indeed, our results cannot be explained simply by a species difference in the level of arousal of the individuals when facing the snake. In fact, if chimpanzees would be overall more aroused and produce more alarm calls in general than bonobos regardless of any socio-cognitive factors, they would call all the time and most individuals would not stop calling once everyone have seen the snake. Our results rather support the idea that bonobos arriving early at the snake produce less alarm call than early arriving chimpanzees possibly because their eagerness to cooperate with the audience is lower than in chimpanzees.

For this reason, I think the authors should cautiously interpret their findings and amend the paper in general, and the discussion accordingly. In particular, they should revise the text throughout to avoid any attribution of higher socio-cognitive skills when using such a paradigm. Here are for instance some paragraphs that would therefore need revisions:

We agree with this point and as mentioned above have now clarified throughout the paper that our intention is not imply second-order intentionality. However, as discussed above, the results do imply some level of complex socio-cognitive skills but as our data address the behaviour not the cognition, we remain agnostic as to what these might be (Line 132 and 425-427).

line 158: “If the interdependence hypothesis holds true, we predict that the more “interdependent” Tai chimpanzees evolved specific cooperation-related cognitive skills (here, keeping track of the behaviour and knowledge status of all present group members). Accordingly, chimpanzees are expected to be better at solving the alarm calling task (i.e. informing all other conspecifics).”

We have rephrased this sentence to imply less active understanding of audience status by the caller to “If the interdependence hypothesis holds true, we predict that the more “interdependent” Tai chimpanzees evolved specific cooperation-related cognitive skills (here, signalling depending on the knowledge status of the audience about the snake). “ (Lines 163-165).

line 273: “In Model 2a we assessed which socio-cognitive parameters affected the general decision to call or not for each individual during the experiment.”

We have rephrased this sentence as: “In Model 2a we assessed which socio-cognitive parameters affected the likelihood to call during the experiment” (Lines 276-277). Since the knowledge level of the audience is a parameter in the analyses and this parameter is socio-cognitive, we would like to keep “socio-cognition” in this sentence.

line 405: "Therefore, the second option seems more likely, that the lower performance of bonobos in our alarm calling task may reflect a difference in intrinsic motivation of individuals to inform everyone (see below) rather than differences in cognitive abilities to track who needs to be informed."

We have rephrased this sentence to: "The second option seems more likely, that the lower performance of bonobos in our alarm calling task may reflect a difference in intrinsic motivation to cooperate by informing others (see below)" (Lines 410-412)

line 415: "Finally, whilst a primary motivation to call in both species seems to be to inform ignorant others"

We would like to keep this phrasing since most apes produce alarm calls only in the presence of ignorant conspecifics which indicates that the primary motivation to call is to produce information to this audience.

line 430: "Non-collaborative cooperative acts (here alarm calling) are expected to be expressed when the benefit for the recipient outweighs the cost for the cooperator [3]. Clearly, potential death following a bite by a Gaboon viper is costlier than the production of alarm calls. Our results indicate that both species appear to have the cognitive capacities to assess the knowledge status of conspecifics."

We have rephrased the last sentence to: "Our results indicate that both species do not differ in the cognitive capacities to call when information is needed by others."

Other minor corrections:

Line 188: '.' Missing after "age categories for both species".

We have added the "." After "age category for both species".

Referee: 4

Comments to the Author(s).

The authors have done an excellent job when revising the manuscript. I have no further questions or objections.

References

1. Dugatkin LA. 1997 *Cooperation Among Animals: An Evolutionary Perspective*. Oxford University Press.
2. Clutton-Brock T. 2009 Cooperation between non-kin in animal societies. *Nature* 462, 51–57. (doi:10.1038/nature08366)
3. Darwin C. 1871 *The Descent of Man and the Selection in Relation to Sex*. London: John Murray.
4. Tomasello M, Melis AP, Tennie C, Wyman E, Herrmann E. 2012 Two Key Steps in the Evolution of Human Cooperation The Interdependence Hypothesis. *Curr. Anthropol.* 53, 673–692. (doi:10.1086/668207)
5. Karakashian SJ, Gyger M, Marler P. 1988 Audience effects on alarm calling in chickens (*Gallus gallus*). *J. Comp. Psychol.* 102, 129–135. (doi:10.1037/0735-7036.102.2.129)
6. le Roux A, Cherry MI, Manser MB. 2008 The audience effect in a facultatively social mammal, the yellow mongoose, *Cynictis penicillata*. *Anim. Behav.* 75, 943–949. (doi:10.1016/j.anbehav.2007.07.014)
7. Wich SA, Sterck EHM. 2003 Possible audience effect in thomas langurs (primates; *presbytis thomasi*): An experimental study on male loud calls in response to a tiger model. *Am. J. Primatol.* 60, 155–159. (doi:10.1002/ajp.10102)
8. Papworth Sarah, Böse Anne-Sophie, Barker Jessica, Schel Anne Marijke, Zuberbühler Klaus. 2008 Male blue monkeys alarm call in response to danger experienced by others. *Biol. Lett.* 4, 472–475. (doi:10.1098/rsbl.2008.0299)